# An Evidence and Consensus-Based Definition of Second Victim: A Strategic Topic in Healthcare Quality, Patient Safety, Person-Centeredness and Human Resource Management

**DOI:** 10.3390/ijerph192416869

**Published:** 2022-12-15

**Authors:** Kris Vanhaecht, Deborah Seys, Sophia Russotto, Reinhard Strametz, José Mira, Sigurbjörg Sigurgeirsdóttir, Albert W. Wu, Kaja Põlluste, Daniela Georgeta Popovici, Raluca Sfetcu, Sule Kurt, Massimiliano Panella

**Affiliations:** 1Leuven Institute for Healthcare Policy, Department of Public Health and Primary Care, KU Leuven, 3000 Leuven, Belgium; 2Department of Quality, University Hospitals Leuven, 3000 Leuven, Belgium; 3Department of Translational Medicine, University of Eastern Piedmont, 28100 Novara, Italy; 4Wiesbaden Business School, RheinMain University of Applied Science, 65183 Wiesbaden, Germany; 5The Foundation for the Promotion of Health and Biomedical Research of Valencia Region, 03550 Alicante, Spain; 6Health Psychology Department, Miguel Hernandez University, 03202 Elche, Spain; 7Department of Political Science, University of Iceland, 102 Reykjavík, Iceland; 8Department of Health Policy and Management, Johns Hopkins Bloomberg School of Public Health, 624 N. Broadway, Baltimore, ML 21205, USA; 9Department of Internal Medicine, Institute of Clinical Medicine, University of Tartu, L. Puusepa 8, 50406 Tartu, Estonia; 10National Institute of Health Services Management, 021253 Bucharest, Romania; 11Nursing Department, Health Sciences Faculty, Karadeniz Technical University, 61080 Trabzon, Turkey

**Keywords:** patient safety, healthcare professionals, second victim, healthcare quality, person-centeredness, human resource management

## Abstract

The concept of second victims (SV) was introduced 20 years ago to draw attention to healthcare professionals involved in patient safety incidents. The objective of this paper is to advance the theoretical conceptualization and to develop a common definition. A literature search was performed in Medline, EMBASE and CINAHL (October 2010 to November 2020). The description of SV was extracted regarding three concepts: (1) involved persons, (2) content of action and (3) impact. Based on these concepts, a definition was proposed and discussed within the ERNST-COST consortium in 2021 and 2022. An international group of experts finalized the definition. In total, 83 publications were reviewed. Based on expert consensus, a second victim was defined as: “Any health care worker, directly or indirectly involved in an unanticipated adverse patient event, unintentional healthcare error, or patient injury and who becomes victimized in the sense that they are also negatively impacted”. The proposed definition can be used to help to reduce the impact of incidents on both healthcare professionals and organizations, thereby indirectly improve healthcare quality, patient safety, person-centeredness and human resource management.

## 1. Introduction

Medical errors due to system flaws and active failures will always occur and there is a continuous need to improve patient safety [1]. One way of improving is by attempting to mitigate the impact of a patient safety incident (PSI) on patients and their family members, who can be referred to by the term first victims. However, in the aftermath of a PSI, healthcare professionals can also suffer. In 2000, Wu introduced the term “second victim” (SV) writing “Many errors are built into existing routines and devices, setting up the unwitting physician and patient for disaster. And, although patients are the first and obvious victims of medical mistakes, doctors are wounded by the same errors: they are the second victims” [2]. Since then, attention to this phenomenon has increased. Nine years later, in 2009, Scott defined an SV as “a health care provider involved in an unanticipated adverse patient event, medical error, and/or a patient related–injury who become victimized in the sense that the provider is traumatized by the event”. Since then, there have been no new contributions to the conceptualization of this phenomenon, though, in practice, the scope of the term SV has been broadened. Frequently, SVs feel personally responsible for the unexpected patient outcomes and feel as though they have failed their patient, second guessing their clinical skills and knowledge base [3]. The prevalence of the SV experiences varies in published reports. For example, during a six-month period after a PSI, this ranged between 9% [4] and 38.7% [5], while SV prevalence was found up to 86.3% for a five-year period [6]. The type of harm also appears to be associated with the prevalence of being an SV. Harrison and colleagues found that 90% of health care professionals reported being involved in at least one near miss of a PSI, with the potential for serious patient harm in 28% and serious patient harm in 17% [7]. A positive effect of the introduction of the term SV is that it raised awareness that healthcare professionals and organizations can also be harmed. Recognition of the phenomenon may help involved parties to cope with an incident and validates the thoughts and emotions they experience. Additionally, for organizations, it provides a gateway towards the cultural changes needed to achieve a patient-centered environment focused on patient safety [8,9]. The large majority of PSIs can be traced back to flawed systemic, strategic, or organizational conditions beyond the influence of individual healthcare workers. In this context, it seems appropriate to regard the involved frontline worker as a second victim [10].

In addition to positive effects of introducing the term SV, there have also been calls to abandon it [8,11,12]. Patient advocates and some physicians are sometimes uncomfortable and dissatisfied with the term [8,11,12]. One major reason given is that it seems to remove the accountability of healthcare professional for practicing safely [11]. Use of the term ‘victim’, could lead healthcare professionals and organizations to believe that patient harm is random, not preventable and caused by bad luck [8]. Although this may be true in some cases, it is clear that both professionals and healthcare organizations need to ensure the safety of patients. By electing sympathy for the healthcare professional, the term SV might encourage passivity and reduce the urgency to address the causes of the PSI [8,11]. It may create the perception that healthcare professionals are just looking after themselves and only think of themselves [12]. In 2017, a group of international experts, including patients, clinicians and healthcare researchers recommended keeping the term second victim as the term is coming into widespread use by clinicians and health care managers as well as policy makers [13].

In 2020, the European Researchers’ Network Working on Second Victims (ERNST) was established by the European Cooperation in Science and Technology (COST) as COST Action 19113. The aim of ERNST is “to facilitate discussion and share scientific knowledge, perspectives, and best practices concerning adverse events in healthcare institutions to implement joint efforts to support SVs and to introduce an open dialogue among stakeholders about the consequences of this phenomenon based on a cross-national collaboration that integrates different disciplines and approaches. For all of this, this network is linked with patient safety issues and the impact of the direct and/or indirectly involved healthcare professional”. One of the core objectives is to further develop the theoretical conceptualization of the SV phenomenon and to develop a common understanding of its definition [14].

During the ERNST meetings, it became clear that the current definitions of Wu (2000) [2] and Scott and colleagues (2009) [3] are in some cases unclear, were both developed in the United States and did not include the current insights. Based on this, there is a need to perform a systematic review to screen the literature for new descriptions or definitions of SV and achieve international consensus about the term SV from healthcare professionals’ view points. Therefore the COST research network on SV (ERNST) posed the following research question: “how is second victim actually defined in the literature and can we define an evidence-based consensus definition on SV?”.

## 2. Materials and Methods

To facilitate the discussion and development of an evidence- and consensus-based definition, A dedicated ERNST working group was launched (WG2). The working group was led by two researchers (KV and MP) with experience of the topic [4,15,16,17,18,19,20,21,22,23,24,25,26,27,28,29,30].

This was a mixed methods study including a systematic review of the published literature between 2010 and 2020, followed by a series of online meetings with an academic task force of ERNST (between September 2020 and April 2022) and a final expert consensus meeting in October 2022.

### 2.1. Information Sources and Search Strategy

Following on two reviews published by Seys et al. [23,30], a literature search was performed using Medline, EMBASE and CINAHL from 1 October 2010 onwards and collected literature until 26 November 2020. The following search strategy was employed: (“second victim,” OR “medical error” OR “adverse event”) AND (“psychology” OR “emotions” OR “feelings” OR “burnout” OR “depression” OR “empathy” OR “attitude of health personnel”) OR “Medical Error” [MeSH] AND “Burnout, Professional” [MeSH] OR “Depressive Disorder” [MeSH] OR “Empathy” [MeSH]. This study was conducted according to the Preferred Reporting Items for Systematic Reviews and Meta-analyses (PRISMA) standard [31]. The intervention (or exposure) and population was a patients safety incident (PSI) involving healthcare professionals. A Patient Safety Incident (PSI) was defined as “an event or circumstance that could have resulted, or did result, in unnecessary harm to a patient” [32].

### 2.2. Eligibility Criteria

Studies were included if any of the following criteria were met. (i) They described a definition or described the term second victims. (ii) They mentioned the prevalence of health care professionals involved in a PSI. The PSI could have occurred during their career or during a well-defined period. (iii) They studied the impact of a PSI on the involved health care professional without restrictions regarding the level of impact. (iv) They assessed what support was provided and/or needed in the aftermath of a PSI.

We excluded studies not published in English, reviews, conference reports, newspaper stories, and anecdotal evidence.

### 2.3. Article Screening

All citations were imported into the citations manager, Endnote X9, and duplicates were initially removed by this citation manager. Duplicates found during the title analyses were manually removed. In the next phase, the titles and abstracts were screened first by two reviewers (DS, KV) to eliminate unrelated studies. Any discrepancy was resolved by two independent investigators (MP, SR). For all remaining relevant articles, the full text was retrieved, and two reviewers examined them independently according to the eligibility criteria.

### 2.4. Data Extraction

Data were extracted from the full text by a single investigator (DS) using a data extraction form. The form included study information (authors, country, year of study), study design, samples (sample size, type of respondents), and outcome measurements (impact of PSI, support in the aftermath of a PSI).

### 2.5. Data Synthesis

The definitions and descriptions of SV found in the literature were examined. Each definition/description was divided into three concepts. These concepts were: (1) who was involved? (2) what happened? and (3) what is the result of what happened? This process was performed by one investigator (DS). A second investigator resolved any questions that occurred during data synthesis (KV).

### 2.6. Building the New Definition

During the first meeting of the Management Committee of ERNST in September 2020, the development of a new definition for SV was assigned to ERNST working group 2. This group consisted of 29 members out of 17 countries and was led by researchers of KU Leuven and Piedmont University (DS, KV, MP and SR). Working group leaders discussed the progress during online meetings every two months. During two online meetings (20–21 April 2021, 21 October 2021) the interim results were discussed within the ERNST consortium. Based on both the academic task force meetings and the ERNST consortium meetings, a new proposed definition was built.

Finally, this definition was discussed within an online expert consensus meeting on 6 April 2022 and finalized in a final consensus meeting in hybrid format on 28 October 2022 in Frankfurt, Germany. Each of the invited experts was a member of the ERNST consortium. In total, nine international experts attended this final consensus meeting of whom eight were from Europe and one from the United States. During all meetings, the literature search, data synthesis and the proposed definition were presented. At the end of each meeting, the experts could give feedback on how to improve the definition and a new definition was built. Each version of the definition was sent by e-mail to all members and they could provide additional suggestions. The version of the definition published in this manuscript was finalized after editing by a native English speaker.

## 3. Results

There were a total of 120,635 titles identified after the initial search. After removing duplicates and title, abstract and full text analysis, 83 studies were included. Figure 1 illustrates the outcome of the search process. Appendix A provides additional details about the included publications.

An overview of the differently used concepts in the definitions or descriptions of SVs is provided in Appendix A. A distinction was made between (1) who is involved, (2) what had happened and (3) result of what had happened. Next, based on the concepts and definitions found in the literature, the following definition was proposed by the academic taskforce to the international experts: “Any health care provider involved in an unanticipated adverse patient event, medical error, and/or a patient related-injury, who was not reckless or malicious, and becomes victimized in the sense that the provider is emotionally impacted by the event. Frequently, second victims feel personally responsible for the unexpected patient outcomes and feel as though they have failed their patient”. The experts proposed several suggestions to improve the definition. The adaptations and the reason for adapting the definition can be found in Table 1.

The proposed definition was accepted unanimously by formal vote of the attending participants of ERNST working group 2 (n = 11) on 28 October 2022 and a final online consensus round with all the authors.

Finally, ERNST concluded the following definition of Second Victim: “Any health care worker, directly or indirectly involved in an unanticipated adverse patient event, unintentional healthcare error, or patient injury, and becomes victimized in the sense that also the worker is negatively impacted” (Box 1).

Box 1New evidence and consensus-based definition of second victim (SV).A second victim is defined as: “Any health care worker, directly or indirectly involved in an unanticipated adverse patient event, unintentional healthcare error, or patient injury, and who becomes victimized in the sense that they are also negatively impacted.”

## 4. Discussion

This is the first study seeking an international definition based on a systematic literature review and internal expert consensus that responds to current conceptualization and interventions to deal with the SV phenomenon. Having an internationally agreed definition has implications for researchers, healthcare organizations developing patient safety plans, professional and patient associations and policymakers. The definition could provide a common basis for training and research. We believe that the future focus of the SV concept should be on quality of care, patient safety, person centeredness and human resource management as this is also the updated vision in a new multidimensional quality model [33] and consistent with the evolving history of quality [25]. This update emphasizes that professionals, patients and kin have an active involvement in quality of care. It also highlights the need for a fluid dialogue between institutions, policymakers, professional and patient groups and loved ones regarding the factors that contribute to patient safety.

This SV definition no longer includes the term “traumatized” as trauma is a diagnosis based standardized classification [34]. Any healthcare worker can become an SV, and there is no need of a medical diagnosis to have this experience. Therefore, we used negatively impacted (in general) rather than traumatized or only emotionally impacted. In the aftermath of a PSI, the healthcare professionals can become involved in a formal complaint or a lawsuit, where the prevalence of a mental health sequelae is higher [27]. In some healthcare professional groups, e.g., radiologists, the risk of malpractice lawsuits and the intense stressed associated with it is increased. This can lead to a certain resistance to disclosing PSI to patients but can also lead to a higher occupational stress of the healthcare professional [35,36,37].

This definition advances the scope of the SV programs in line with recent approaches that have incorporated professionals who are negatively affected in their professional work by their concern that a patient does not achieve the expected outcome of the intervention/treatment he/she undergoes.

Regarding the type of PSI [38], our definition excludes injury because of reckless or malicious behavior. However violation of safe procedures should be differentiated from intended reckless behavior and unintended negligence or desperate decisions. An example of the latter is a healthcare worker who needs to act in an acute situation despite a lack of necessary resources, who consciously takes a risk. Healthcare organizations should realize that in most cases system failures contribute the most to patient safety incidents. Sometimes the workers repeatedly violated the rules because they had no alternatives and want to deliver good care. It is either taking the risk of possible failure or not taking care of the patient. This behavior can be referred to as a “work-around” and generally indicates one or more system flaws. In these cases, if a patient is harmed, the workers are most likely to also be harmed. On the other hand, our definition clearly excludes reckless or malicious healthcare professionals, whom injured patients consider it most inappropriate to think of as SV.

With this new evidence and consensus based definition, we hope that more colleagues and healthcare organizations recognize the SV experience, are more able to speak-up about them, and to take actions to provide optimal support [30]. A lesson for healthcare organizations should be realizing that these SVs, feeling bad and very guilty, having reduced confidence in their own abilities, may cause further harm to many future patients, further reducing safety and quality of healthcare [22]. For this reason alone, organizations should support their SVs. However, they are also obligated to learn from the event, e.g., with morbidity and mortality conferences or root cause analysis, and take action to prevent the event from happening again.

There have been impassioned calls to abandon the term SV [8,11,12]. We understand why some stakeholders might find this concept offensive but adverse events occur because of both active failures and latent organizational conditions. The worker becomes victimized because of this. Therefore, it is necessary that we put both first and second victims in the perspective of patient safety. We support the conclusions of the 2017 paper on this issue [13]. A clear, international consensus and an inclusive definition of SV will help to further clarify the debate. It is important to highlight that a professional who becomes an SV and is not provided with support runs an increased risk of being involved in additional safety incidents.

A strength of this study is that the definition is evidence based literature and includes the consensus of international experts working in different countries using different protocols and procedures, and it works in different cultures and legal and policy systems. A limitation of this study is that we did not conduct a systematic evaluation of the quality of the included articles to ensure inclusion of any available concepts that were previously used for defining SV. In addition, the concept of the third victims, i.e., the involved healthcare organization, was not part of this definition. This will need to be developed in the future. This updated definition of SV is written from the point of view of academic and healthcare professionals. The literature on the viewpoints of the involved patients and relatives, known as first victims after adverse events, was frequently discussed during the working group meetings.

ERNST will conduct further discussions of this new definition with patient advocacy groups and their kin. Further research will need to evaluate the acceptance of this new consensus definition in different cultures and policy environments by all stakeholders including first, second and third victims.

## 5. Conclusions

Our study defined an SV as “Any health care worker, directly or indirectly involved in an unanticipated adverse patient event, unintentional healthcare error, or patient injury, and becomes victimized in the sense that also the worker is negatively impacted”. This definition is the result of systematically analyzing the different descriptions and definitions of SV and includes current conceptualization, approaches and scopes of possible interventions. To ensure the inclusion of any available concept that was previously used for defining SV, all articles were included without performing a systematic evaluation of the quality. An international group of experts finalized the consensus definition. By this evidence and consensus based definition, all stakeholders will be empowered to recognize the second victims concept and further enhance research, training, and actions to support patient safety initiatives to continuously improve the care that patients receive.

## Figures and Tables

**Figure 1 ijerph-19-16869-f001:**
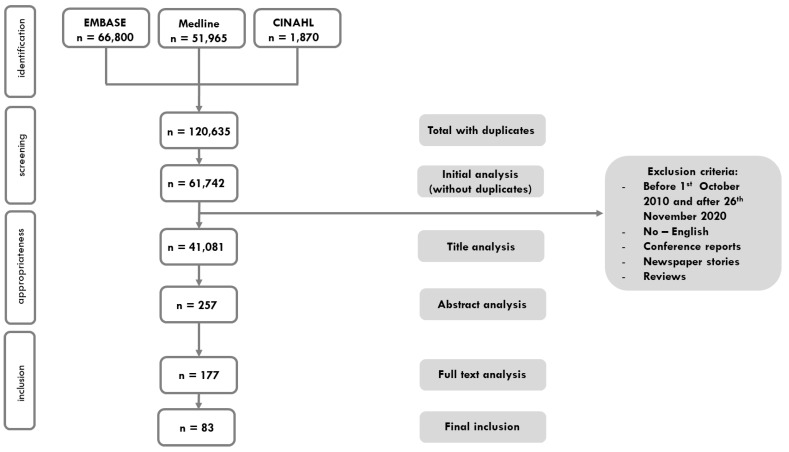
PRISMA flow diagram.

**Table 1 ijerph-19-16869-t001:** Adaptations to the definition of second victim (SV) based on the experts.

Words in Proposed Definition by Academic Taskforce KULeuven-Piemonte University	Words in Final Definition Based on International Expert Consensus Meeting	Explanation of the Adaptation
Any healthcare provider	Any healthcare worker, directly or indirectly involved …	The event can lead to a negative impact on any type of professional (clinical or non clinical) that was directly involved in an active failure or who was the victim of a latent organizational condition. However, colleagues who were indirectly involved or even not involved in the event itself can have a negative impact. Think about colleagues who were working on the same ward or team during the event, or even non-clinical managers or quality improvement facilitators who feel responsible for the latent condition.
unanticipated adverse patient event, medical error, and/or a patient related–injury	… in an unanticipated adverse patient event, unintentional healthcare error, or patient injury …	If a healthcare worker is involved in something potentially catastrophic but with a good outcome, they can also become an SV. Next to this, near misses or events where the clinician had no active failure but there was a negative outcome, should be included. We add “unintentional” as the act was not reckless or malicious and broaden to healthcare error as not all errors are medical.
Healthcare provider	… and who becomes victimized in the sense that …	It could be a food service worker, a health facilities worker, a cleaner, who becomes victimized (because of active failures and/or latent conditions) … as long as there is in some way direct or indirect contact with the patient they may feel affected. This is also the case for students.
Frequently, second victims feel personally responsible for the unexpected patient outcomes and feel as though they have failed their patient.	Removed	This sentence is a clarification and should not be part of a definition.
the healthcare provider	… they are also … (=healthcare worker is also)	We emphasize that patients and kin are the first victims. By “also” we state it is not only the healthcare worker who is negatively impacted. With this we acknowledge the primary impact of the incident on patients and kin.
emotionally impacted	… negatively impacted.	It can be any type of negative impact, not only emotional. We acknowledge that there can also be a positive impact of the incident (learning curve, more attention, …).
by the event	Removed	This is a repeat and therefore removed in the consensus definition.

## Data Availability

The data presented in this study are available on request from the corresponding author.

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
