# Peer review of "An Evidence and Consensus-Based Definition of Second Victim: A Strategic Topic in Healthcare Quality, Patient Safety, Person-Centeredness and Human Resource Management"

_ijerph, 2022, doi:10.3390/ijerph192416869_

Round 1
Reviewer 1 Report
The Abstract of the article contains all the required information, however it does not follow the MDPI guidelines. Authors should remove the terms "Methods:"; "Results:" and "Conclusion:". Although this is the indication of other publishers, it is not at all the recommendations of MDPI!
It is not at all clear why the authors chose as Keywords: "systematic/scoping review" and "consensus methods". It is not clear why a member of the scientific community who is interested in accessing this article would choose these keywords, since any research in any area of research can use them in their research and investigation methodology. It is recommended to the authors to remove these two keywords!
It would be very beneficial for the article, if the authors included the "Research Question" in the article! What motivated the authors to develop this research? What was the gap that the authors identified, which led them to conduct this research? This indication in the article would help to improve the article.
The conclusions are very vague and should be reviewed and strengthened by the authors. It would be better if the authors put here some of the most important values obtained with this research, and reinforce in this way the importance and contribution of this research to the society and scientific community.
It would be very important that the authors at the end of the conclusions indicate the limitations of this research. All research has its limitations, and they are a natural part of research. However, the authors' sharing of their limitations helps the scientific community to understand the research and some of the results obtained.
Author Response
Dear reviewer,
We thank you for the careful reading of the paper and your suggestion on our manuscript. Please find our item-by-item response below. Changes are marked with track changes in the manuscript. The mentioned line numbers refer to the document with track changes.
Comment 1: The Abstract of the article contains all the required information, however it does not follow the MDPI guidelines. Authors should remove the terms "Methods:"; "Results:" and "Conclusion:". Although this is the indication of other publishers, it is not at all the recommendations of MDPI!
Answer 1: Thank you for informing us that we needed to remove the headers in the abstract (line 34, 36, 41 and 44). We have changed the manuscript according to the guidelines.
Comment 2: It is not at all clear why the authors chose as Keywords: "systematic/scoping review" and "consensus methods". It is not clear why a member of the scientific community who is interested in accessing this article would choose these keywords, since any research in any area of research can use them in their research and investigation methodology. It is recommended to the authors to remove these two keywords!
Answer 2: We fully agree with your comment and removed these two keywords from our manuscript (line 50).
Comment 3: It would be very beneficial for the article, if the authors included the "Research Question" in the article! What motivated the authors to develop this research? What was the gap that the authors identified, which led them to conduct this research? This indication in the article would help to improve the article.
Answer 3:
In 2020 the European Researchers’ Network Working on Second Victims (ERNST) was established by the European Cooperation in Science and Technology (COST). The aim of ERNST is “to facilitate discussion and share scientific knowledge, perspectives, and best practices concerning adverse events in healthcare institutions to implement joint efforts to support SVs and to introduce an open dialogue among stakeholders about the consequences of this phenomenon based on a cross-national collaboration that integrates different disciplines and approaches.”. Wu introduced SV in 2000 and in 2009 Scott and colleagues expanded the definition of SV. The term SV has been widespread used by not only healthcare professionals but also healthcare managers and policy makers. During the meetings with the ERNST working groups it became clear that the current definitions of Wu (2000) and Scott et al. (2009) are in some cases unclear, were both developed in the United States and did not includes the current insights. There is need to perform a systematic review to screen the literature for new descriptions or definitions of SV and achieve international consensus about the term SV from healthcare professionals’ point of view. Therefore the COST research network on SV (ERNST) posed the research following question: “how is second victim actually defined in the literature and can we define an evidence-based consensus definition on SV?”
The motivation to develop this research was made more clear in the introduction and the two research questions were also added (lines 108-115).
Comment 4: The conclusions are very vague and should be reviewed and strengthened by the authors. It would be better if the authors put here some of the most important values obtained with this research, and reinforce in this way the importance and contribution of this research to the society and scientific community.
Answer 4: This is a very interesting point. We revised the manuscript according to your suggestion. We now started our conclusion section with our definition which is the main outcome of this study (lines 286-289).
Comment 5: It would be very important that the authors at the end of the conclusions indicate the limitations of this research. All research has its limitations, and they are a natural part of research. However, the authors' sharing of their limitations helps the scientific community to understand the research and some of the results obtained.
Answer 5: We agree that each research has their strengths and limitations. The strength and limitations were mentioned at the end of the discussion section (lines 270 – 280). In the final conclusion we added a sentence to put our results better in perspective (lines 292 – 294).
Kind regards
Reviewer 2 Report
Dear,
Sorry for the delay, I intend to deliver the next revision on the agreed date.
Article
“An evidence and consensus-based definition of second victim. A strategic topic in healthcare quality, patient safety, person-centredness and human resource management.”
The article brings an important debate for patients and health professionals. It is very well written, clear methodology and achieves its objectives. The article can be accept in present form.
Author Response
Dear reviewer,
We thank you for the careful reading of the paper and for the positive comments on our manuscript.
Kind regards
Round 2
Reviewer 1 Report
The article has been improved by the authors!